# Deep Incomplete Multi-View Semi-Supervised Multi-Label Learning Network with Unbiased Loss

Quanjiang Li
National University of Defense
Technology
Changsha, China
liquanjiang@nudt.edu.cn

Tingjin Luo*
National University of Defense
Technology
Changsha, China
tingjinluo@hotmail.com

Mingdie Jiang
National University of Defense
Technology
Changsha, Country
jiangmingdie20@nudt.edu.cn

Jiahui Liao
National University of Defense
Technology
Changsha, China
liaojiahui19@nudt.edu.cn

Zhangqi Jiang
National University of Defense
Technology
Changsha, China
jiangzq@nudt.edu.cn

## Abstract

Due to the explosive growth in data sources and label categories, multi-view multi-label learning has garnered widespread attention. However, multi-view multi-label data often exhibits incomplete features and a huge number of unlabeled instances, due to the technical limitations and high cost of manual labeling in practice. Learning for such simultaneous missing of view features and labels is crucial but rarely studied, particularly when the labeled samples are limited. In this paper, we tackle this problem by proposing a novel Deep Incomplete Multi-View Semi-Supervised Multi-Label Learning method (DIMvSML). Specifically, to improve high-level representations of missing features, deep graph network is firstly employed to recover the feature information with structural similarity relations. Meanwhile, we design the structure-specific deep feature extractors to obtain discriminative information and preserve the cross-view consistency for the recovered data with instance-level contrastive loss. Furthermore, to eliminate the bias of the estimate of the risk that the semi-supervised multi-label methods minimise, we design a safe estimate framework with an unbiased loss and improve its empirical performance by using pseudo-labels of unlabeled data. Besides, we provide both the theoretical proof of better estimate variance and the intuitive explanation of our debiased framework. Finally, extensive experimental results on public datasets validate the superiority of DIMvSML compared with state-of-the-art methods.

## CCS Concepts

• **Computing methodologies → Neural networks**; **Semi-supervised learning settings**.

*Corresponding author

## Keywords

Deep Learning, Incomplete Multi-view Learning, Semi-supervised Classification, Multi-label Learning.

**ACM Reference Format:**

Quanjiang Li, Tingjin Luo, Mingdie Jiang, Jiahui Liao, and Zhangqi Jiang. 2024. Deep Incomplete Multi-View Semi-Supervised Multi-Label Learning Network with Unbiased Loss. In *Proceedings of the 32nd ACM International Conference on Multimedia (MM '24), October 28-November 1, 2024, Melbourne, VIC, Australia.* ACM, New York, NY, USA, 9 pages. https://doi.org/10.1145/3664647.3681414

## 1 Introduction

Multi-label learning has attracted increasing attention due to its widespread application in areas such as text classification [3, 34], image annotation [2, 31], and computer vision tasks [7, 10]. Furthermore, with the exponential increase of data sources and feature extraction methods, it is no longer adequate to describe and analyze instances from a singular perspective [8]. In real-world applications, objects are usually processed in multiple views, like face information captured by diverse sensors and image data stored using both video and audio techniques. Doubtlessly, the utilization of multi-view data enables comprehensive and accurate description of observed instances [9, 45]. Besides, multi-view data offers abundant data presentation modes, which can be combined with multi-label to represent the rich information content and semantic structure of complex data [39, 44]. Therefore, this paper focuses on the multi-view multi-label classification task, namely MVMLC.

For MVMLC, many methods have been proposed, such as the manifold regularization MVMLC [25], potential semantic-aware LSA-MML [41] and label-embedding based method [46]. However, these traditional methods assume that the given data has complete views and labels, which is violated in practice. On the one hand, the heterogeneous data collected from multiple sources may contain missing views due to the quality of storage equipment and the difficulty of storage methods [4]. For instance, in multi-view multimedia annotation tasks, video, audio and subtitle serve as distinct views. It is common to face situations where not all multimedia content encompasses all three views [37]. On the other hand, manual tagging of all labels being both challenging and expensive,

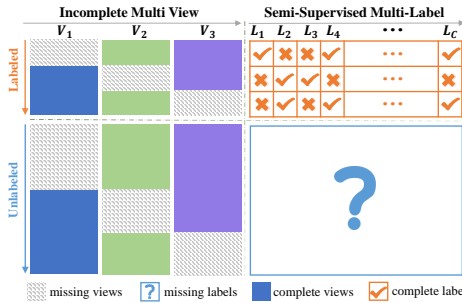

**Figure 1: The example of incomplete multi-view semi-supervised multi-label data.**

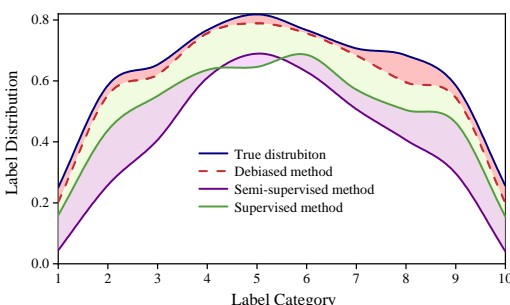

**Figure 2: The label distribution obtained by three different methods on Yeast.**

label information in real datasets frequently exhibits varying degrees of incompleteness [23]. Clearly, the absence of views and labels detrimentally impacts MvMLC. In recent years, the challenges posed by the two types of missing have received widespread attention and some works which can simultaneously handle both issues have been proposed. iMvWL [33] combined two weighted matrix factorization models into a unified framework to extract the consensus representation and derive a subspace from missing views with multiple labels. NAIML [17] exploited both consensus across multiple views and the global and local structures among multiple labels from rank constraint. By incorporating missing information into the weighted fusion and classification module, the proposed DD-IMvMLC [37] could effectively explore available data and label information to obtain the discriminative feature extractor and classifier. DICNet [21] emphasized the utilization of stacked auto-encoders to exploit the high-level semantic representations of samples. Besides, it introduced an incomplete instance-level contrastive learning to capture consistent representations. Wen et al. [22] proposed LMVCAT, which employed two transformer-style modules for cross-view feature aggregation and multi-label classification and utilized an adaptively weighted view fusion module to obtain view-consistent embedding features.

These MVMLC methods under incomplete data only focus on the partial absence of multiple labels and each instance has a subset of labels that can be utilized to infer the missing ones. However, multi-view data not only suffers from missing features but also encounters unlabeled instances in reality. For example, the features and labels of tumor patient data are obtained from various examinations and tests [27]. Certain patients may face limitations in undergoing MRI or PET scans, or in providing blood samples for laboratory testing, thereby leading to corresponding views being unavailable. Besides, factors such as research focus, resource constraint, and annotation complexity greatly contribute to patient labels, including tumor types, grading, treatment responses and so on, remaining unannotated. In aggregate, real-world datasets often present simultaneous problems of missing views and numerous unlabeled instances as shown in Fig. 1. There are few methods available today that can effectively address both issues. We are all aware that in supervised multi-label scenarios, correlations within labels are beneficial for label recovery. However, in semi-supervised problem [24, 32], where a sample lacks any annotations, solely considering methods for handling missing from label relevance is simplistic. In fact, to address

the problem of semi-supervised multi-label under incomplete views, we need to focus on the following three aspects: i) Since the stability of the model will be seriously affected by the limited availability of data, we should consider maximizing the reconstruction of missing information. Rather than establishing losses only on observed data [38, 47], we should enhance feature semantic from a data recovery perspective and maintain the stability of the subsequent modules [16]. ii) High-level representations should be explored to improve the contribution of features to classification with limited labeled instances. iii) Improperly incorporating unlabeled data to construct an unsupervised loss undermines the unbiased estimation of ideal risk from supervised losses, which makes the algorithm often lack the support of statistical theory [35, 43]. Besides, it introduces bias to the solution of supervised losses optimization, resulting in potential performance degradation of semi-supervised methods. In Fig. 2, we select ten relatively balanced labels of Yeast to perform the experiment of label distribution simulation and only set 20% of the test data to be labeled. We can observe that when introducing unlabeled data and learning with traditional biased semi-supervised losses, the performance (marked in purple) tends to be worse than using only supervised data (marked in green). Moreover, the harm caused by the bias expands as the number of label categories increases, which needs to be controlled in multi-label learning. Therefore, the third aspect is to improve the framework of loss functions to make semi-supervised model safe and robust.

To tackle these problems, we propose a novel deep incomplete multi-view semi-supervised multi-label learning method named DIMvSML. Specifically, for mitigating the negative influence of missing views, DIMvSML employs the Graph Neural Network to recover the missing data by capitalizing on the existing similarity relations. Based on the auto-encoder structures, we design feature extractors and decoder networks to learn high-level semantic and discriminative representations from all views. In addition, to preserve the cross-view consensus, we adopt the instance-level contrastive loss to enhance the mutual information between different views. An unbiased version of loss function is designed to eliminate the risk of latent downgrade due to the introduction of the unlabeled data and we prove theoretically that this framework also has a lower estimate variance. As depicted in Fig. 2, utilizing our unbiased loss function leads to stable performance compared to the traditional semi-supervised loss when using unlabeled data. During training, the pseudo-labels are assigned to explore the additional

supervisory information contained in unlabeled data. The main contributions of our work are summarised as follows:

- We propose the DIMvSML to solve this crucial, but rarely studied problem. To our knowledge, this is the first GNN-based multi-view multi-label learning framework capable of handling both incomplete views and few labeled instances.
- DIMvSML is a unified framework designed to recover the absent views, preserve the high-level semantic representations with cross-view consensus and provide a safe risk estimation framework simultaneously.
- Extensive experimental results present that our DIMvSML outperforms other compared approaches in almost all cases, demonstrating its superiority and effectiveness.

## 2 Methodology

**Notations and Problem Formulation.** Suppose an incomplete multi-view dataset with $N$ instances and $V$ views, i.e., $\mathcal{X} = \{X^{(v)}\}_{v=1}^{V}$, where $X^{(v)} = \{x_i^{(v)}\}_{i=1}^{N} \in \mathbb{R}^{N \times d_v}$ is the $d_v$ dimensional feature matrix of the $v$-th view. Let $Y \in \{0, 1\}^{N \times C}$ represent the label matrix and $C$ is the number of categories. Besides, $y_i \in \{0, 1\}^{C}$ is a row vector that denotes the label of the $i$-th instance. $M \in \mathbb{R}^{N \times V}$ is an indicator matrix, where $M_{i,j} = 1$ indicates that the $i$-th instance has the feature of the $j$-th view, otherwise $M_{i,j} = 0$ means the feature is missing and set as 'NaN'. For convenience, we define $n_l$ and $n_u$ as the number of labeled and unlabeled instances, which satisfies $n_l \ll N$ and $n_l + n_u = N$. We also denote $\mathcal{L}$ and $\mathcal{U}$ as the index spaces for labeled and unlabeled instances, respectively.

Obviously, the missing features will affect the learning of complementary and consistency across all views and prevent subsequent modules from using valid information. Furthermore, the scarcity of labeled samples will seriously limit the learning of multi-label semantics, which demands better utilization of unlabeled data to prevent the harm caused by the bias of the semi-supervised loss. To address these challenges, we propose a novel deep learning framework named DIMvSML. The main framework of our DIMvSML is illustrated in Fig. 3. Specifically, DIMvSML consists of three main modules: (a) GNN-based feature completion module for recovering the incomplete feature information; (b) Multi-view representation learning module for capturing the high-level semantic representations and discriminative information from all views; (c) Safe semi-supervised multi-label learning module for providing an unbiased semi-supervised risk estimator with lower variance.

### 2.1 GNN-based Feature Completion Module

Since the absence of features will lead to the poor performance of deep learning [30], data recovery is required to realize data augmentation. Recently, GNN-based approaches have garnered attention in data recovery owing to their capacity to extract the geometric details embedded within data [19, 36]. Sato [28] further presented theoretical evidence that substantiates the effectiveness of GNNs in recovering hidden features. Therefore, we employ GNNs to recover the missing views by leveraging the similarity relations between the available data.

Firstly, we construct the view-specific graph $S^{(v)} \in \mathbb{R}^{N \times N}$ through $k$-nearest neighbors ($k$-NN) algorithm. $S^{(v)}$ demonstrates

the relations between the corresponding instances of existing data in the $v$-th view, where $S_{i,j}^{(v)} = 1$ means $M_{i,v} M_{j,v} = 1$ and $x_j^{(v)}$ is the neighbor of $x_i^{(v)}$. Considering the consistency across multiple views, similarity relations between instances in existing views are valid for the missing views. Therefore, we transfer the established graph relations to find the available instances that are associated with the missing ones in each view. Then the transferred $k$-NN graph can be obtained by

$$K^{(v)} = \sum_{k=1, k \neq v}^{V} S^{(k)} \, \text{diag} \left( M_{:,v} \right), \tag{1}$$

where operator $\text{diag}(\cdot)$ forms a diagonal matrix, and $M_{:,v}$ denotes the $v$-th column of the matrix $M$. Secondly, we employ $K^{(v)}$ as the adjacency matrix and the related existing features as the iuput nodes in each view-specific GNN to recover the missing data. After the propagation of relation information over $K^{(v)}$ in the first layer of the GNN, the initially reconstructed data can be obtained by

$$\hat{x}_i^{(v)} = \sigma \left( b_v + \sum_{K_{i,j}^{(v)} \geq 1} K_{i,j}^{(v)} \omega_v x_j^{(v)} \right), \tag{2}$$

where $b_v$ and $\omega_v$ denote the bias and transformation matrix of the $v$-th view, respectively. In our experiments, we set $\sigma$ as the rectified linear unit (ReLU) activation function. Finally, we combine the reconstructed missing data with the existing data to acquire the recovered matrices $\{\widetilde{X}^{(v)}\}_{v=1}^{V}$. Moreover, in order to consolidate the recovery performance, we minimize the rebuilding loss $L_{\text{rb}}$ only over recovered data as follows:

$$L_{\text{rb}} = \sum_{v=1}^{V} \sum_{K_{i,j}^{(v)} \geq 1} \left\| \widetilde{X}_{i,:}^{(v)} - x_j^{(v)} \right\|_2^2 (1 - M_{i,v}). \tag{3}$$

### 2.2 Multi-view Representation Learning Module

Due to the small number of labeled instances in semi-supervised learning (SSL), we need an efficient representation module to simultaneously explore the high-level semantic of features, unique characteristic for each view and substantial connections between features and labels. Therefore, we adopt the deep neural network rather than the shallow linear model for adaptively extracting the advanced representations. Besides, considering that different views have their distinctive characteristics, the feature extraction network $E^{(v)}(\cdot)$ and decoder network $D^{(v)}(\cdot)$ should be tailored for each view. Following [37, 40], we adopt the well-known network structure of stacked auto-encoder. These view-specific networks are all composed of multi-layer perceptrons but with different hidden layer dimensions. By constructing such network structure, we can effectively capture the discriminative information inherent in each view. Therefore, we set the structure of both the $E^{(v)}(\cdot)$ and $D^{(v)}(\cdot)$ as four stacked linear layers with ReLU activation functions i.e., {Linear, ReLU, Linear, ReLU, Linear, ReLU, Linear}. Specifically, for the $d_v$ dimensional feature data of the $v$-th view, the dimensions of the four linear layers in the encoder network are adaptively set as $0.8d_v$, $0.8d_v$, 1500, and $d$, where $d$ is the corresponding dimension of the last layer and can be adjusted according to the number of label categories. In reverse, the dimensions of the decoder network

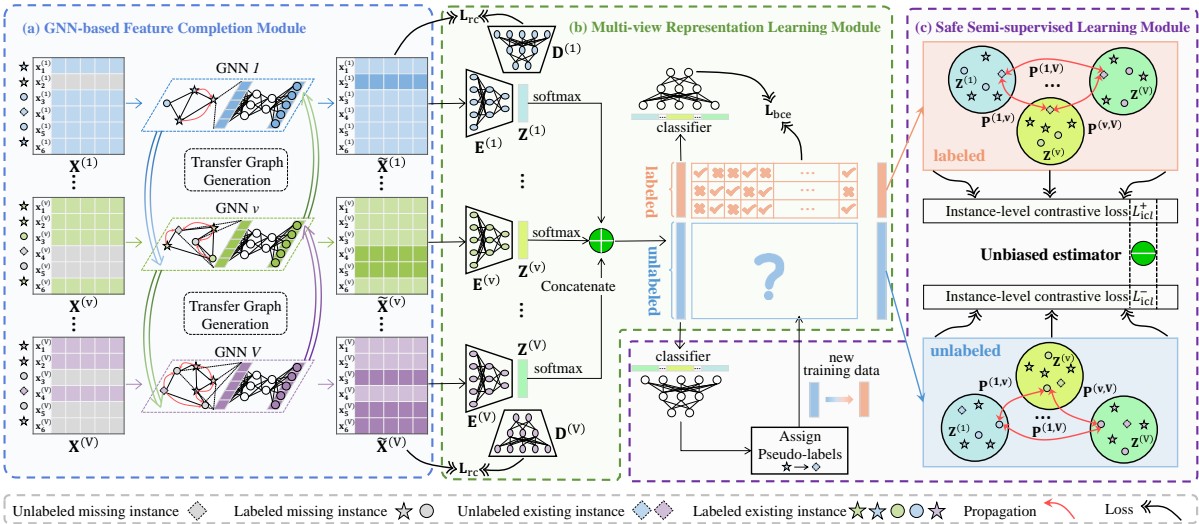

**Figure 3: The main framework of our proposed DIMvSML, which is composed of three modules: (a) GNN-based feature completion module; (b) Multi-view representation learning module and (c) Safe semi-supervised learning module.**

are set as 1500, $0.8d_v$, $0.8d_v$, $d_v$. Through extraction of networks, we can get the representation matrix $Z^{(v)} = E^{(v)}(\widetilde{X})$ by minimizing

$$L_{\text{rc}} = \frac{1}{n_l V} \sum_{v=1}^{V} \left[ \sum_{i \in \mathcal{L}} \left( \left\| \widetilde{X}_{i,:}^{(v)} - D^{(v)} \left( Z_{i,:}^{(v)} \right) \right\|_2^2 / d_v \right) \right]. \quad (4)$$

To integrate the coded features, we obtain a common representation matrix $Z \in \mathbb{R}^{dV \times N}$ by concatenating the view-specific representations, , which maximizes the retention of recovered features while minimizing complexity. For the purpose of effectively exploring the correlations between features and labels, we employ deep network classifier which transforms the feature space into a probability space associated with the labels, i.e., the elements of the output can be regarded as the probability of the instance to the corresponding label. Therefore, we design the classifier as the combination of two stacked linear layers and a Sigmoid activation function, where the dimensions are set as $d/2$ and $C$. For improving prediction accuracy, we use the cross entropy [5] to guide model training. Suppose $F \in \mathbb{R}^{N \times C}$ is a prediction matrix generated by classifier, we employ the following loss for labeled instances:

$$L_{\text{bce}} = -\frac{1}{n_l C} \sum_{i \in \mathcal{L}} \sum_{j=1}^{C} \left( Y_{ij} \log \left( F_{ij} \right) + \left( 1 - Y_{ij} \right) \log \left( 1 - F_{ij} \right) \right). \quad (5)$$

### 2.3 Safe Semi-supervised Learning Module

To prevent the introduction of unlabeled data from causing performance degradation, we provide a safe semi-supervised loss function framework in this section. Assume that the shared parameters of the whole network is $\Theta$ and the ultimate objective of our training framework is to minimise a ideal risk $\mathcal{R}$ over a data distribution $p(x, y)$. Since the distribution $p(x, y)$ is unknowable, we generally minimise a empirical risk $\hat{\mathcal{R}}(\Theta)$, which acts as a surrogate for $\mathcal{R}$ and is computed on a sample of $N$ i.i.d points drawn from $p(x, y)$. Considering supervised losses from each labeled instance, we suppose

$L_{\text{rc}} = \frac{1}{n_l} \sum_{i \in \mathcal{L}} G(\Theta; x_i)$ and $L_{\text{bce}} = \frac{1}{n_l} \sum_{i \in \mathcal{L}} T(\Theta; x_i; y_i)$. Besides, we let loss $L(\Theta; x_i; y_i) = G(\Theta; x_i) + \lambda_1 T(\Theta; x_i; y_i)$. Then the supervised risk to minimise is

$$\hat{\mathcal{R}}_{CC}(\Theta) = \frac{1}{n_l} \sum_{i \in \mathcal{L}} L(\Theta; x_i; y_i). \quad (6)$$

This traditional supervised risk estimate is unbiased and converges wisely to $\mathcal{R}(\Theta)$. However, a notable limitation of this framework under semi-supervised problem is that a considerable amount of unlabeled data is not utilized. Therefore, we employ the instance-level contrastive loss [20] on the unlabeled data to maximize the mutual information between the representations of different views. To calculate the mutual information, we use a Softmax activation function $\sigma_s$ at the last layer of the encoder and then we obtain that $\widetilde{z}_i^{(v)} = \sigma_s(z_i^{(v)})$, which is treated as a distribution probability vector [14]. In other words, $\widetilde{z}^{(v)}$ and $\widetilde{z}^{(v^*)}$ ($1 \le v < v^* \le V$) can be seen as the distribution of two discrete cluster assignment variables over $d$ classes. Therefore, we can compute the joint probability distribution as below:

$$P^{(v,v^*)} = \frac{1}{n_u} \sum_{i \in \mathcal{U}} \left( \widetilde{z}_i^{(v^*)} \right)^T \widetilde{z}_i^{(v)}. \quad (7)$$

Then the mutual information between the $v$-th and $v^*$-th view can be calculated through

$$\ell_{v,v^*} = -\sum_{t=1}^{m} \sum_{t'=1}^{m} P_{t,t'}^{(v,v^*)} \ln \left( \frac{P_{t,t'}^{(v,v^*)}}{\left( P_t^{(v)} \right)^{\alpha+1} \left( P_{t'}^{(v^*)} \right)^{\alpha+1}} \right), \quad (8)$$

where $P^{(v)}$ and $P^{(v^*)}$ are the marginal probability distribution of the $v$-th and $v^*$-th view. In our experiments, we simply fix the balanced parameter $\alpha$ to 9. The loss $L_{icl}^-$ under unlabeled data can be obtained by enumerating the mutual information between

different views, i.e.,

$$L_{\mathrm{icl}}^{-} = \frac{1}{V} \sum_{1 \le v < v^* \le V} \ell_{v,v^*}. \qquad (9)$$

To facilitate subsequent analysis, we turn $L_{\mathrm{icl}}^{-}$ into the form $L_{\mathrm{icl}}^{-} = \frac{1}{n_u} \sum_{i \in \mathcal{U}} H(\Theta; x_i)$. The concrete form of the three instance-level losses $G(\Theta; x_i)$, $T(\Theta; x_i; y_i)$ and $H(\Theta; x_i)$ is easily acquired and will be listed in Appendix. After introducing unlabeled data , we aim to minimise the SSL risk :

$$\hat{\mathcal{R}}_{SSL}(\Theta) = \frac{1}{n_l} \sum_{i \in \mathcal{L}} L(\Theta; x_i; y_i) + \frac{\lambda_2}{n_u} \sum_{i \in \mathcal{U}} H(\Theta; x_i). \qquad (10)$$

Since unlabeled data lacks the necessary labeled guidance for classification, introducing it often carries the risk of potential performance degradation, especially when the data distribution assumption is not satisfied [18]. Even though the learning methods presented in the Eq. (10) can handle some basic SSL problems, the SSL risk estimate is biased or even asymptotic, which not only hinders the use of statistical learning theory, but also damages the actual effect of the model [26]. Moreover, the biased harm becomes prominent when the number of label categories increases. Hence, in multi-label classification, we should compensate for this bias in the loss function. Inspired by Hugo et al. [29], we obtain the following unbiased version of the SSL estimator:

$$\hat{\mathcal{R}}_{DeSSL}(\Theta) = \frac{1}{n_l} \sum_{i \in \mathcal{L}} L(\Theta; x_i; y_i) + \frac{\lambda_2}{n_u} \sum_{i \in \mathcal{U}} H(\Theta; x_i) - \frac{\lambda_2}{n_l} \sum_{i \in \mathcal{L}} H(\Theta; x_i). \qquad (11)$$

This framework uses labelled data to annul the bias, which do not rely on data distribution assumption. In addition to being unbiased, this framework also has favorable estimate variance. To measure the variance of the risk estimate in the Eq.(11), we require information regarding instance tagging. Therefore, we introduce a binary random variable $r \sim \mathcal{B}(\pi)$ that states whether or not a data point is labelled. $r_i = 1$ denotes the $i$-th instance is labeled and $r_i = 0$ denotes missing. $\pi \in (0, 1)$ is the probability of being labelled. Under the assumption that the missingness of a label is independent of its feature and value, we can obtain the following theorem:

THEOREM 1. *When $\lambda_1$ is fixed, the function $\lambda_2 \mapsto \mathbb{V}\left(\hat{\mathcal{R}}_{DeSSL}(\Theta) \mid r\right)$ reaches its minimum for:*

$$\lambda_2{}^* = \frac{n_u}{n} \left( \frac{\mathrm{Cov}(G(\Theta; x, y), H(\Theta; x))}{\mathbb{V}(H(\Theta; x))} + \frac{\mathrm{Cov}(\lambda_1 T(\Theta; x, y), H(\Theta; x))}{\mathbb{V}(H(\Theta; x))} \right)$$

*and at $\lambda_2{}^*$:*

$$\mathbb{V}\left(\hat{\mathcal{R}}_{DeSSL}(\Theta) \mid r\right)\Big|_{\lambda_2{}^*} = \left(1 - \frac{n_u}{n} \rho_{L,H}^2\right) \mathbb{V}\left(\hat{\mathcal{R}}_{CC}(\Theta) \mid r\right)$$
$$\le \mathbb{V}\left(\hat{\mathcal{R}}_{CC}(\Theta) \mid r\right),$$

*where $\rho_{L,H} = \mathrm{Corr}(L(\Theta; x, y), H(\Theta; x))$.*

A detailed proof of this theorem is presented in Appendix. From the theorem 1, we can know that the variance of the unbiased estimate $\hat{\mathcal{R}}_{DeSSL}(\theta)$ is less than that of $\hat{\mathcal{R}}_{CC}(\Theta)$ using only supervised data when $\lambda_1$ and $\lambda_2$ meet certain conditions. This theorem also guides us to simultaneously adjust $\lambda_1$ and $\lambda_2$ to achieve a stable risk estimation effect in practice. When our estimate is unbiased and the variance is smaller, we can theoretically ensure that our semi-supervised module is safe when introducing the unlabeled data and no worse than using only supervised data. To validate

the correctness of our unbiased framework analysis, we train our DIMvSML on Yeast and split the test dataset into 20% labeled and 80% unlabeled data to calculate the $\hat{\mathcal{R}}_{SSL}(\Theta)$ and $\hat{\mathcal{R}}_{DeSSL}(\Theta)$ risks that we compared to the oracle risk estimate using all the test set. For variance test experiment, we split 50 times the test set to estimate the variance of the risk estimator. Besides, we compute $\lambda_2{}^*$ using the entire test set. As shown in Fig. 4, the result illustrates that $\hat{\mathcal{R}}_{DeSSL}(\Theta)$ is unbiased for any value of $\lambda_2$ and its variance can be optimised by adjusting $\lambda_2$ when $\lambda_1$ is fixed. Besides, it can be seen that $\mathbb{V}(\hat{\mathcal{R}}_{DeSSL}(\Theta))$ is less than $\mathbb{V}(\hat{\mathcal{R}}_{SSL}(\Theta))$ in most cases and the theoretical value of $\lambda_2{}^*$ is close to the minimum point calculated from the actual sampling.

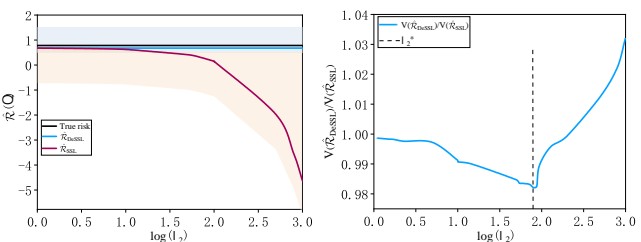

Figure 4: Intuitive explanation of our theoretical analysis. (Left) Risk estimate value for $\hat{\mathcal{R}}_{SSL}(\Theta)$ and $\hat{\mathcal{R}}_{DeSSL}(\Theta)$ compared to the true value of the risk. (Right) The influence of $\lambda_2$ on the raios of $\mathbb{V}(\hat{\mathcal{R}}_{DSSL}(\Theta))\mathbb{V}(\hat{\mathcal{R}}_{SSL}(\Theta))$ when $\lambda_1 = 1$.

Due to the capability to supply new labeled data for training, pseudo-label methods have gained significant prominence in deep semi-supervised classification tasks [15]. Therefore, we assign pseudo-labels to exploit additional supervisory information and improve model performance. Since the classification confidence exhibited in the early stage of classifier is low, we choose the output of classifier as labels for unlabeled instances at half of the total training epoch. Then all instances are incorporated into training process.

## 2.4 Training Strategy

The training strategy employed in DIMvSML contains two phases: pre-training and alternative optimization. In the pre-training phase, we only use the rebuilding loss $L_{\mathrm{rb}}$ to simply train the view-specific GNNs. During the alternate optimization phase, the three proposed modules mutually complement each other, resulting in simultaneous enhancement of classification performance. Denote the loss $L_{\mathrm{icl}}^{+} = \frac{1}{n_l} \sum_{i \in \mathcal{L}} H(\Theta; x_i)$, the overall loss of our DIMvSML in alternative optimization phase be formulated as

$$L = L_{\mathrm{rc}} + \lambda_1 L_{\mathrm{bce}} + \lambda_2 (L_{\mathrm{icl}}^{-} - L_{\mathrm{icl}}^{+}), \qquad (12)$$

where $\lambda_1$ and $\lambda_2$ are penalty coefficients. The training process of our DIMvSML is summarized in Appendix.

## 3 Experiments

### 3.1 Experimental Setup

**Datasets.** In our experiments, six public multi-view multi-label datasets are selected to validate the proposed method, i.e., **Yeast** [12], **Corel 5k** [6], **VOC 2007** [7], **Esp Game** [1], **IAPR TC-12**

**Table 1: Ranking Loss, ACC, AP and AUC of different methods on six public datasets with LER fixed to 20% and PER fixed to 50%. The best result on each row is bolded and the second-best result is underlined.**

| Dataset | Metric | TM3L | iMVWL | NAIML | DD-IMvMLC | DICNET | LMVCAT | DIMvSML |
|---------|--------|------|-------|-------|-----------|--------|--------|---------|
| Yeast | Ranking Loss↓ | 25.21±0.55 | 21.86±0.68 | 23.67±0.80 | 25.63±4.89 | 22.52±0.49 | 24.54±0.78 | **20.04±0.64** |
| | ACC ↑ | 69.53±0.31 | 72.13±0.40 | 75.53±0.34 | 71.37±2.16 | 70.46±1.02 | 73.51±0.73 | **78.34±0.72** |
| | AP↑ | 66.73±0.57 | 69.92±0.31 | 69.26±0.73 | 67.72±3.75 | 69.32±0.51 | 67.57±0.99 | **72.48±0.41** |
| | AUC↑ | 55.42±1.19 | 49.99±0.76 | 59.09±1.19 | 52.42±2.06 | 54.30±1.06 | 60.41±1.45 | **62.47±1.58** |
| Corel 5k | Ranking Loss↓ | 30.39±0.32 | 22.62±0.77 | 22.59±0.42 | 16.93±0.44 | 16.40±0.35 | 17.33±0.79 | **14.42±0.90** |
| | ACC ↑ | 98.65±0.02 | 97.46±0.09 | 98.63±0.03 | 98.68±0.00 | 98.69±0.01 | 98.69±0.01 | **98.70±0.01** |
| | AP↑ | 22.53±0.75 | 14.74±1.96 | 27.61±0.64 | 26.17±0.45 | 28.54±0.74 | 26.97±0.87 | **29.90±1.79** |
| | AUC↑ | 58.53±1.31 | 50.18±0.74 | 61.67±0.63 | 54.22±0.83 | 55.80±1.07 | 56.62±1.70 | **66.38±2.83** |
| VOC 2007 | Ranking Loss↓ | 29.05±0.41 | 32.54±1.24 | 29.36±1.72 | 29.76±2.36 | 24.67±0.41 | 20.57±1.38 | **17.87±0.65** |
| | ACC ↑ | 92.29±0.04 | 87.83±0.25 | 88.37±2.45 | 92.48±0.04 | 92.68±0.05 | 91.22±0.55 | **93.41±0.09** |
| | AP↑ | 44.44±0.48 | 41.02±0.71 | 41.84±1.00 | 43.28±1.06 | 46.45±0.45 | 51.21±1.32 | **56.05±0.57** |
| | AUC↑ | 59.60±0.20 | 50.08±0.14 | 49.99±1.33 | 52.33±1.01 | 58.36±0.84 | 73.45±1.40 | **76.30±1.42** |
| Esp Game | Ranking Loss↓ | 31.76±0.29 | 25.28±0.61 | 24.13±0.33 | 21.75±1.61 | 19.98±0.25 | 19.45±0.26 | **18.26±0.32** |
| | ACC ↑ | 98.15±0.02 | 96.90±0.09 | 98.25±0.00 | 98.24±0.00 | 98.24±0.01 | 98.24±0.01 | **98.26±0.01** |
| | AP↑ | 18.02±0.17 | 15.76±2.13 | 25.03±0.26 | 21.56±1.35 | 25.00±0.35 | 26.04±0.28 | **26.17±0.56** |
| | AUC↑ | 55.82±0.45 | 49.92±0.18 | 58.94±0.49 | 52.69±1.85 | 53.98±0.18 | 62.00±0.89 | **65.10±0.58** |
| IAPR TC-12 | Ranking Loss↓ | 25.40±0.19 | 23.62±0.80 | 24.63±0.57 | 19.72±0.99 | 17.16±0.23 | 16.50±0.42 | **15.21±1.69** |
| | ACC ↑ | 97.90±0.02 | 96.61±0.06 | 97.04±1.22 | 98.03±0.01 | 98.04±0.01 | 98.02±0.01 | **98.06±0.01** |
| | AP↑ | 22.53±0.19 | 16.50±1.23 | 19.88±0.23 | 22.26±0.63 | 25.18±0.16 | 27.11±0.57 | **27.90±2.35** |
| | AUC↑ | 59.18±0.28 | 49.93±0.38 | 50.25±0.45 | 55.48±1.14 | 59.20±0.16 | 65.13±0.94 | **69.04±3.64** |
| MIR FLICKR | Ranking Loss↓ | 24.14±0.24 | 19.95±0.29 | 19.89±2.95 | 17.12±1.10 | 15.30±0.34 | 14.09±0.77 | **13.83±0.25** |
| | ACC ↑ | 86.90±0.12 | 83.87±0.05 | 84.35±0.98 | 87.69±0.03 | 87.58±0.07 | 87.61±0.64 | **88.78±0.17** |
| | AP↑ | 47.55±0.44 | 44.41±0.43 | 45.75±1.37 | 50.71±1.69 | 54.29±0.55 | 55.46±0.09 | **57.56±0.45** |
| | AUC↑ | 58.30±0.28 | 50.13±0.30 | 49.43±0.90 | 60.94±2.92 | 63.19±0.43 | 72.38±1.14 | **74.36±0.30** |

[11], **MIR FLICKR** [13]. For the first dateset, we pick Genetic Expression and Phylogenetic Profile as two views; for the other five datasets, we choose six types of features as six views, i.e., GIST, HSV, DenseHue, DenseSift, RGB, and LAB.

**Comparison Methods.** To validate the effectiveness of DIMvSML, we compare it with six state-of-the-art approaches, which can be categorized into two groups: traditional methods and deep methods. Traditional methods include: TM3L [42], iMvWL [33], NAIML [17], while deep methods include DD-IMvMLC [37], DICNet [21], LMVCAT [22]. Five of them are introduced in the preliminaries and TM3L is a multi-view multi-label classification method, which can handle partial multi-label data. Noting that except for TM3L, the other five methods can handle both feature and label missing simultaneously. Therefore, the missing views are populated by their average instance calculated from the corresponding available instances of the same view for TM3L in our experiment. For all comparison methods, we will prioritize the parameter settings recommended in the original code implementations or specified in their respective papers.

**Data Preparation.** Each dataset can be divided into training, validation and test sets in the ratio of 7:1:2. To simulate the partial view setting, we randomly remove some views of samples from each set. Concretely, according to the pre-set partial example ratio (PER), PER% instances are randomly selected as incomplete instances, which randomly missing $1 \sim V-1$ views (at least one view per instance is available to keep the total number of samples constant). For the SSL situation, according to the pre-set labeled example ratio

(LER), we randomly select LER% instances as labeled instances in the training dataset.

**Implementation Details.** The $k$-NN graphs are constructed based on the Euclidean distance metric, where the neighbor number $k$ is fixed to 10 for all datasets. The Adam optimizer is employed with an initial learning rate of 0.0001 for optimizing the training loss. In addition, Ranking Loss (RL), Accuracy (ACC), Average Precision (AP) and adapted area under curve (AUC) are adopted as four evaluation metrics. All the experimental results are derived from ten independent runs of the methods, and the final average results along with their corresponding standard deviations are presented. Our model is implemented by PyTorch on one NVIDIA GeForce RTX 4090 with GPU of 24GB memory.

### 3.2 Performance Evaluation

To comprehensively verify our DIMvSML, we compare it with six competitive methods from two key aspects: i) view missing and ii) label insufficient. For view missing, we fix LER to 20%, while PER is selected in {0%, 10%, 30%, 50%, 70%, 90%}. For label insufficient, we fix PER to 50%, while LER is chosen in {15%, 20%, 25%, 30%, 35%, 40% 45%}. The statistical results are presented in Table 1, Fig. 5 and Fig. 6. Tabel 1 displays the four metrics with LER fixed at 20% and PER fixed at 50%, while Fig. 5 and Fig. 6 show the AUC when LER and PER change respectively. The additional results of Rankingloss and AP are shown in Appendix.

Regarding the missing view, we can find that 1) When PER= 0%, DIMvSML achieves the best performance on all datasets, which

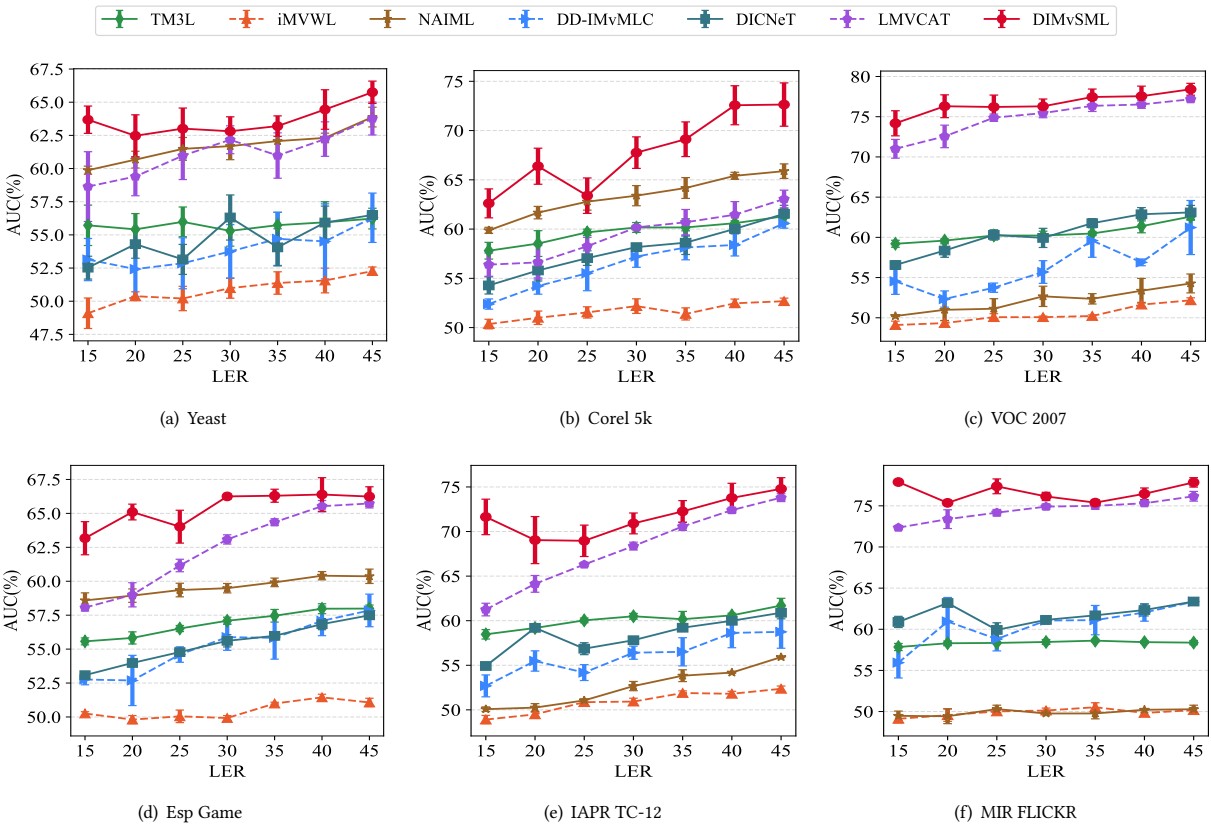

**Figure 5: AUC comparisons on six datasets with LER varying from** 15% **to** 55% **while PER**=50%.

indicates that the proposed method is also stable and effective in the classification tasks under complete views. 2) With PER increasing from 10% to 90%, DIMvSML still outperforms the other six methods. It shows our method adequately addresses the problem of missing views and is beneficial for feature completion. 3) DIMvSML has the capability to address scenarios where the absence of certain views significantly impacts the classification process. In other words, even when all compared methods fail to deliver satisfactory results, our approach continues to exhibit a considerable effect. For example, DIMvSML and the most competitive method achieve AUC of 70.27% and 61.78% when PER=10% on Corel 5k, revealing an increase of nearly 10% percent. As for label insufficient, we have the following observations: 1) Our method achieves better among all compared methods in almost all cases. 2) Our DIMvSML is robust to few labeled instances since it consistently exhibits relatively promising performance with lower LER. For example, DIMvSML and the second-best method LMVCAT achieve AUC of 74.79% and 73.8% when LER=45% on IAPR TC-12. As LER=15%, the performance of DIMvSML is 71.64% and superior to 61.25% of LMVCAT.

### 3.3 Ablation Study

The ablation experiments on VOC 2007 and IAPR TC-12 are carried out to thoroughly investigate the impact of the three critical modules of DIMvSML. When the GNN-based feature completion

module ($S_1$) is disabled, we employ the average strategy to fill the missing data. When the Multi-view representation learning module ($S_2$) is disabled, we simply concatenate each view and remove the loss $L_{\mathrm{rc}}$. As for the safe semi- supervised learning module, we compare the classification performance under the $\hat{\mathcal{R}}_{CC}(\Theta)$ and the $\hat{\mathcal{R}}_{DeSSL}(\Theta)$ loss framework since we focus on whether the introduction of unlabeled data would actually improve performance. The ablation results are listed in Table 2. We can know that: 1) In the first three rows, performance is reduced when $S_1$ and $S_2$ are removed respectively, which indicates that our method is effective for data recovery and feature extraction. 2) In the last two rows, the performance of our debiased framework $\hat{\mathcal{R}}_{DeSSL}(\Theta)$ is better than $\hat{\mathcal{R}}_{CC}(\Theta)$ using only supervised data. It demonstrates that our method indeed enhances model performance with the introduction of unlabeled data and provide a reliable effect for semi-supervised classification.

### 3.4 Parameter Sensitivity

We conduct experiments on VOC 2007 and IAPR TC-12 to analyze the sensitivity of $\lambda_1$ and $\lambda_2$. Two parameters are selected from the range of $\{0.01, 0.1, 1, 10, 100\}$ and the joint influence are presented in the heatmap as shown in the Fig. 7. Since the difference between the best performance and the worst is 22.9 on VOC 2007, we can learn that our method is sensitive to both parameters. The result further

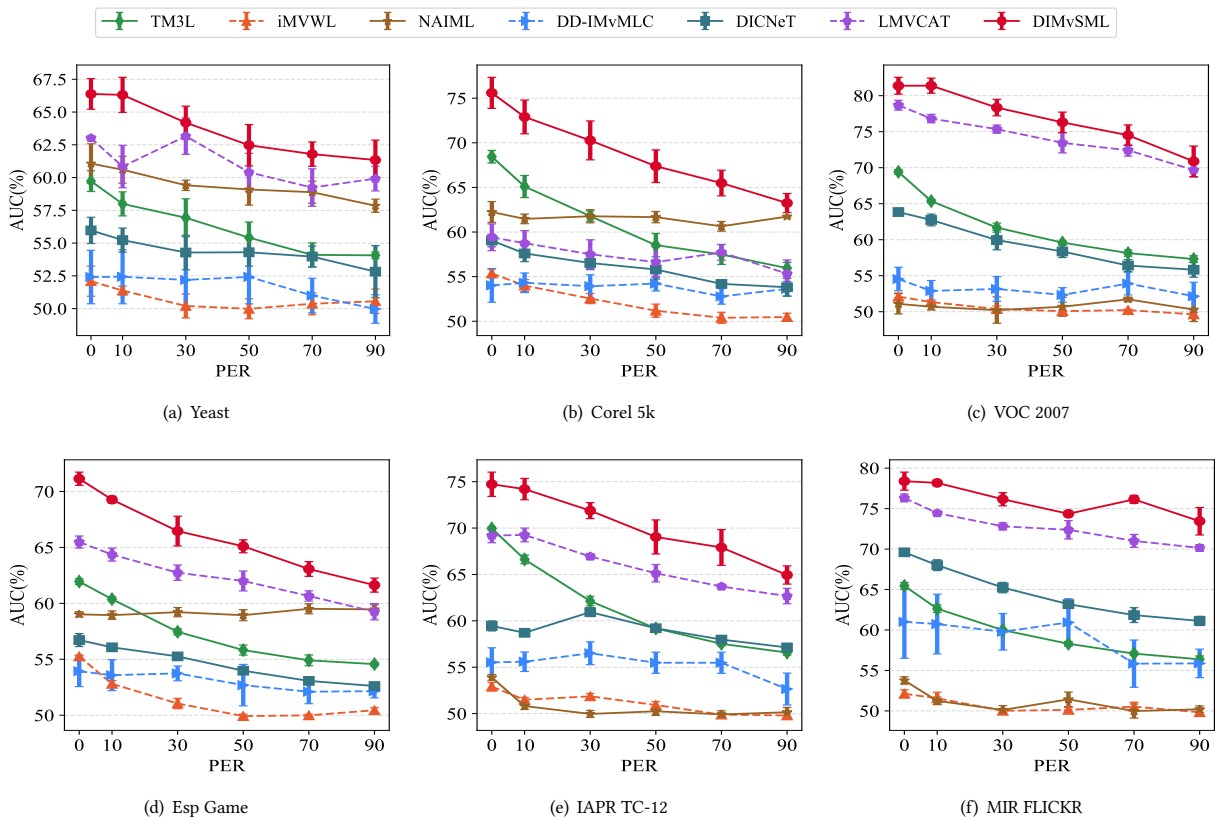

Figure 6: AUC comparisons on six datasets with PER varying from 0% to 90% while LER=20%.

Table 2: Ablation study on VOC 2007 and IAPR TC-12 with PER=50% and LER=20%. '✓' and '✗' represent the used and not used corresponding item, respectively.

| $S_1$ | $S_2$ | $\hat{\mathcal{R}}_{CC}(\Theta)$ | $\hat{\mathcal{R}}_{DeSSL}(\Theta)$ | VOC 2007 | | IAPR TC-12 | |
|---|---|---|---|---|---|---|---|
| | | | | AP | AUC | AP | AUC |
| ✗ | ✓ | ✗ | ✓ | 54.14 | 73.81 | 25.98 | 67.15 |
| ✓ | ✗ | ✗ | ✓ | 49.08 | 68.98 | 19.93 | 59.92 |
| ✓ | ✓ | ✗ | ✓ | **56.05** | **76.30** | **27.90** | **69.04** |
| ✓ | ✓ | ✓ | ✗ | 53.55 | 73.28 | 23.54 | 66.21 |
| ✓ | ✓ | ✗ | ✓ | **56.05** | **76.30** | **27.90** | **69.04** |

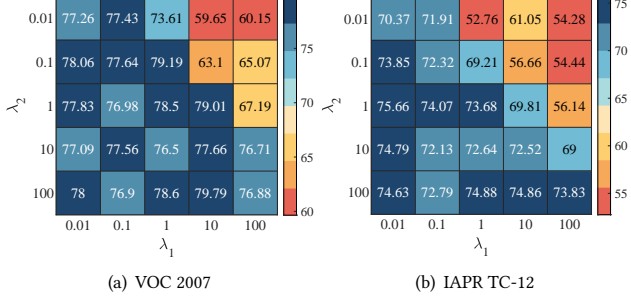

(a) VOC 2007      (b) IAPR TC-12

Figure 7: Parameter analysis of the trade-off parameters $\lambda_1$ and $\lambda_2$ on VOC 2007 and IAPR TC-12.

validates the Theorem 1 and emphasizes the need to simultaneously adjust $\lambda_1$ and $\lambda_2$ to approach the condition for the Theorem 1, which ensures stable performance.

## 4 Conclusion

To tackle the incomplete multi-view semi-supervised multi-label problem, we propose a novel deep learning based method named DIMvSML in this paper. DIMvSML incorporates both the GNN-based feature completion, view-specific representation extraction network and safe semi-supervised multi-label learning module to preserve discriminative feature and enhance the semantic label

information. Besides, we design an unbiased loss to alleviate the bias from large amount of unlabeled data and provide theoretical analysis of our safe risk estimator. Therefore, our DIMvSML can eliminate the negative effect of the incomplete data and use unlabeled information safely for efficient classification. Finally, extensive experimental results on six public datasets demonstrate the effectiveness and superiority of DIMvSML. In the future, we will further extend to solve other multi-label problems under incomplete views, such as class-imbalance and noisy labels etc.

## Acknowledgments

This work was supported by the National Science Foundation of China Grant [62036013, 62376281], and the NSF for Huxiang Young Talents Program of Hunan Province under Grant [2021RC3070].

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
