# OpenReview forum: "Deep Incomplete Multi-View Network Semi-Supervised Multi-Label Learning with Unbiased Loss"
_acmmm.org/ACMMM/2024/Conference — MM2024 Poster_

### Official Review · Reviewer_3YoP · 2024-05-23

**Rating:** 5
**Confidence:** 4

**Summary:**

This paper proposes a deep learning framework, DIMvSML, for addressing the common issue of incomplete views and labels in multi-view multi-label data applications. By integrating graph neural networks, the model enhances the accuracy of feature recovery. Additionally, through providing unbiased risk estimation, it not only optimizes label prediction accuracy but also improves the model's stability in solving semi-supervised problems. Experimental results demonstrate that DIMvSML outperforms existing state-of-the-art techniques.

**Strengths:**

In this paper, they conduct the analyses on both methodological and theoretical levels. Methodologically, DIMvSML constructs the model based on GNN networks and unbiased risk estimation losses. Theoretically, it demonstrates that the unbiased loss framework adopted exhibits superior statistical properties. The experimental results including comparisons under various experimental settings, ablation studies, and sensitivity analysis of parameters are provided to a comprehensive analysis of the algorithm's performance.

**Limitations:**

1. As we know, the problem of multi-label and multi-view data with simultaneous missing views and a large number of unlabeled samples addressed in this paper is very challenging. Is there a real and practical scenario corresponding to this problem?

2. In the introduction section, the author conducts a label distribution simulation experiment on the Yeast dataset and demonstrates that the debiased method outperforms the traditional semi-supervised paradigm. However, it is known that label distribution does not equate to the final prediction accuracy. It is possible to achieve good simulation results by predicting all zeros. Can the author further clarify whether this label distribution simulation experiment can convincingly demonstrate the effectiveness of the debiased framework proposed in this paper, and why?

3. This paper proposes the use of mutual information in the contrastive loss to learn consistency across different views. However, the author does not further explain the source of the mutual information calculation or why it is presented in the form described in the article.

**Suitability:**

3

---

### Official Review · Reviewer_qhet · 2024-05-23

**Rating:** 5
**Confidence:** 4

**Summary:**

This paper introduces a novel Deep Incomplete Multi-View Semi-Supervised Multi-Label Learning method (DIMvSML), addressing the problems of incomplete features and missing labels in multi-view multi-label data. DIMvSML employs graph neural networks to recover missing feature information and designs structure-specific deep feature extractors to capture discriminative information while maintaining cross-view consistency. Additionally, an unbiased loss function is introduced to improve the risk estimation of semi-supervised multi-label methods. Extensive experimental results demonstrate the superiority of DIMvSML over state-of-the-art methods.

**Strengths:**

1. To address the problem, this paper employs graph neural networks to recover views and utilizes a debiased loss function to enhance the performance of the semi-supervised multi-label learning, And the complementary and consistency across different views are considered in this paper.

2. Experimental results under various feature missing ratios and label annotation ratios validate the effectiveness of the proposed method.

**Limitations:**

1. DIMvSML uses graph neural networks to recover the information of incomplete multi-view features. What are the benefits of using GNNs compared to other feature recovery methods, and what is the principle behind its ability to recover features?

2. Please clarify the meaning of Eq. (8), and explain how the joint distribution between different views and their marginal distributions are computed in contrastive learning.

3. This paper mentions the strategy of adding pseudo-labels to unlabeled samples to explore additional supervisory information. However, it does not verify the correctness of these pseudo-labels. In scenarios with a large number of label categories, pseudo-labels are likely to be less accurate. Does the presence of incorrect pseudo-labels introduce more noise during network training, potentially affecting the model's performance?

**Suitability:**

3

---

### Official Review · Reviewer_SHwM · 2024-05-24

**Rating:** 3
**Confidence:** 4

**Summary:**

In this paper, the authors propose a novel solution with the deep incomplete multi-view semi-supervised multi-label learning method. The method employs graph neural networks to recover feature information through structural similarity relations. Additionally, they design structure-specific deep feature extractors and maintain cross-view consistency for the recovered data using instance-level contrastive loss. Furthermore, to address the bias in risk estimation minimized by semi-supervised multi-label methods, they introduce a safe risk estimation framework with an unbiased loss, improving empirical performance with pseudo-labels of unlabeled data.

**Strengths:**

1. The authors creatively use graph neural networks to address the problem of incomplete multi-view semi-supervised multi-label learning, providing a new perspective on multi-view learning.

2. A security risk estimation method is presented, with rigorous mathematical proofs ensuring the method's stability on unlabeled data.

3. Extensive experiments demonstrate the effectiveness of the method.

**Limitations:**

1. The paper appears to have some typos, such as in the method names on lines 224 and 787.

2. The structure of the paper is not sufficiently clear. The implementation details and experimental analysis discussed at the end of Sections 2.2 and 2.3, respectively, would be more appropriately placed in the "Experiments" section.

3. The experimental analysis in the paper is not sufficiently in-depth. For example, in Section 3.4, the authors only conclude that the method is sensitive to parameters. I am also curious why in Figure 5, DIMvSML seems unrelated to changes in LER, while other methods generally increase with increasing LER.

4. In the multi-view representation learning module, the authors fuse the representations of different views by concatenation. Could this approach make it difficult to capture the relationships between views? Have other methods been tried, such as addition, setting learnable parameters, or using attention mechanisms?

**Suitability:**

3

---

### Official Review · Reviewer_JyRB · 2024-05-25

**Rating:** 5
**Confidence:** 4

**Summary:**

This paper introduces the Deep Incomplete Multi-View Semi-Supervised Multi-Label Learning (DIMvSML) approach to solve the challenges posed by incomplete features and limited labels in multi-view multi-label learning. Specifically, DIMvSML leverages graph neural networks to reconstruct missing view data and employs an unbiased loss to optimize risk estimation. Comprehensive experimental results demonstrate the superior performance of this method.

**Strengths:**

1. Novelty: The authors designed a deep learning model proposed to recover missing views and obtain unbiased risk estimation for multi-label semi-supervised learning.
2. Validation: It includes theoretical analysis of the model and its parameters.
3. Experimental: The experiments demonstrate the method's performance from multiple perspectives, including comparative experiments as well as ablation study and convergence.

**Limitations:**

1. This paper employs a GNN network to initially restore missing features, followed by leveraging feature representation and contrastive learning techniques. Since subsequent modules hinge on complete features, the restoration process must precede all other operations. Therefore, a detailed explanation of the network's end-to-end training process is warranted, encompassing the various training steps, the utilization of distinct loss functions at different stages, and the precise point at which pseudo-labels are integrated.
2. The paper introduces a novel concept of debiasing in the context of multi-label semi-supervised learning. While traditional semi-supervised methods may still yield satisfactory results without explicit debiasing, the author should elaborate on the underlying reasons for this phenomenon. Additionally, further insight into the risks associated with neglecting debiasing could provide a deeper understanding of its importance.
3. When it comes to the optimization of the network model, this paper lacks crucial details. Specifically, it remains unclear whether mini-batch training is adopted during each epoch and, if so, how batches comprising solely labeled or unlabeled samples are handled. These details are crucial for understanding the network's learning process and potential biases.

**Suitability:**

3

---

### Meta-Review · Area_Chair_SVaP · 2024-07-01

**Recommendation:** Accept (Poster)
**Confidence:** 5

**Metareview:**

This paper studies semi-supervised multi-view multi-label learning and focuses on the challenging cases with missing views and missing labels. In this paper, the authors proposed a novel network, DIMvSML, which addresses the missing views issue by feature recovery and designs a safe risk estimate framework with an unbiased loss to address the incomplete label learning issue. After the rebuttal phase, four reviewers reach a consistent recommendation to accept the paper. The final recommendation for the paper is acceptance.